# Diaryl Bismuthides and Acyl Bismuthanes Enable Visible-Light-Induced Reversible Carbon Monoxide Insertion and Extrusion

Felix Geist [1], Benedikt Narz[1], Darian Boguslawski[1], Tobias Dunaj [1], Sascha Reith [1], Jordi Poater [2,3] & Crispin Lichtenberg [1] ✉

Carbon monoxide (CO) is one of the simplest and most fundamental molecules that has fascinated chemists for decades. Early-on, chemists have recognized and exploited its favorable properties as a key reagent in large-scale metallurgical processes, productions of base- and fine chemicals, and (more recently) potential medical applications. Most of these transformations rely on reactions that proceed in the coordination sphere of transition metals, with reversible reactions playing a crucial role in catalytic transformations. Recent endeavors have brought main group elements on the stage of carbonyl chemistry (carbonyl = metal-bound CO), but reversible reactions (especially insertion and extrusion reactions) are very rare and remain limited to examples of thermal initiation. Here we show that an innovative access to acyl-bismuth compounds, $R_2Bi-C(O)R'$, can be granted in a stepwise redox approach. The target compounds could be isolated and fully characterized. As a unique feature, these complexes enable the visible-light-driven reversible CO extrusion and insertion of carbon monoxide in the coordination sphere of a main group metal.

Carbon monoxide (CO) is a diatomic molecule that represents an iconic ligand in organometallic chemistry and a reagent of paramount importance in metallurgical processes[1,2] and catalysis[3]. The (reversible) coordination and insertion of carbon monoxide (Fig. 1a) plays a crucial role in many industrial processes, e.g. the Monsanto and Cativa™ processes for the manufacture of acetic acid[4–6], the Fischer-Tropsch process where syngas (CO/H₂) is converted to hydrocarbons[7–11], and the Mond process for the purification of nickel[2,12]. Additionally, the controlled release of carbon monoxide (decarbonylation) has found applications in bond formation processes[13–15] and therapeutic applications[16–19]. These processes have traditionally been dominated by transition metal compounds, but in recent years, researchers have developed innovative approaches to bring selective transformations of CO into the realm of molecular main group chemistry[20–22]. Successful strategies have been based on low-valent and multiply-bonded compounds, geometrically constrained species, and frustrated Lewis pairs. Initial findings cover diverse types of transformations with carbon monoxide, including the fundamentally important reactions of coordination/dissociation and insertion/extrusion (Fig. 1b). The detailed understanding and control of these elementary reactions in the chemistry of carbon monoxide with main group elements is the basis for the development of selective and competitive transformations. Consequently, the identification of model systems that can equilibrate between isolable compounds via coordination/dissociation and insertion/extrusion pathways is extremely important. For the coordination/dissociation of CO, a small number of such systems has been identified[20,21], but the situation is even more challenging for the case of insertion/extrusion pathways. Very rare examples include one low-valent boron compound[23] and an aluminum species that reversibly incorporates CO into a cyclic framework[24]. As a crucial point, the shuttling between the insertion and the extrusion of CO in these two cases does not cover the synthetically important class of compounds

¹Department of Chemistry Philipps-Universität Marburg, Hans-Meerwein-Str. 4, Marburg, Germany. ²Departament de Química Inorgànica i Orgànica & IQTCUB, Universitat de Barcelona, Barcelona, Spain. ³ICREA, Barcelona, Spain. ✉e-mail: crispin.lichtenberg@chemie.uni-marburg.de

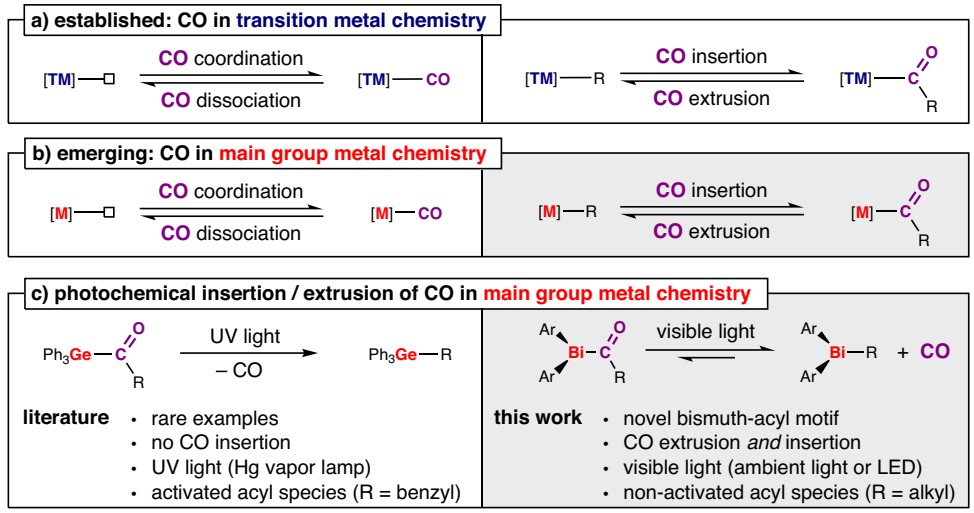

**Fig. 1 | Reactivity of carbon monoxide (CO) in transition and main group metal chemistry. a** Fundamental transformations of carbon monoxide in transition metal chemistry and **b** main group metal chemistry. **c** State of the art in photochemically-driven CO insertion/extrusion with main group complexes. UV = ultraviolet, Ph = phenyl, R = organic substituent, Ar = aryl, LED = light-emitting diode.

with terminal acyl groups C(O)R (R = alkyl, aryl) and is achieved through conventional thermal activation of the reactants. In an orthogonal approach, the light-driven extrusion of CO from a terminal acyl germane has been reported, but required activated acyl groups and very harsh reaction conditions (400 W high-pressure Hg vapor lamp, containing "hard" UV-C radiation which is not part of ambient light as it is absorbed by the ozone layer)[25]. In addition, no signs of reversibility of this transformation have been reported, a crucial part en route to potential catalytic transformations of CO. In view of the light-responsive nature of many organobismuth species[26–28], bismuth acyl compounds are extremely attractive targets, but have remained elusive to date, despite attempts at their synthesis[29].

Here we report the synthesis of isolable and thoroughly characterized diaryl bismuthides, [BiR$_2$]$^-$, which enable an inverse-polarity-approach to access bismuth acyl compounds (thereby introducing the Bi–C(O)R structural motif to main group chemistry) and engage in visible-light-induced reversible CO extrusion/insertion reactions.

## Results and Discussion

The in-situ generation of simple bismuthides [BiR$_2$]$^-$ has been suggested based on trapping experiments (R = aryl, alkyl)[30–36]. These parent compounds do not benefit from stabilizing electronic effects (such as aromatization or the α-silyl effect)[37–40] that aim at significant delocalization of electron density. One set of single-crystal X-ray diffraction data of a diphenyl bismuthide (without additional analytical data or a yield) has serendipitously been reported[41], granting first glimpses at the yet elusive class of diaryl bismuthides. But a rational synthesis facilitating the isolation and characterization of these simple bismuthides has not been reported to date. We envisioned that the reduction of isolable, low-valent dibismuthanes, R$_2$Bi–BiR$_2$, might pave the way to the elusive parent bismuthides [BiR$_2$]$^-$. Indeed, the reaction of compounds **1-4** with potassium graphite (KC$_8$) in the presence of [2.2.2]cryptand (crypt) in THF gave the target compounds **5-8** in 38-84% isolated yield (Fig. 2). The bismuthides are intensely dark orange to dark red compounds that appear almost black in the solid state and could be fully characterized.

$^1$H and $^{13}$C{$^1$H} NMR spectra of compounds **5-8** show the expected signal pattern for the respective aryl substituents at bismuth and one equivalent of [2.2.2]cryptand chelating a potassium ion. Extreme changes in chemical shifts of the aryl groups compared to the neutral parent compounds (BiR$_3$) and the dibismuthane starting materials are not apparent[42–48], suggesting a significant localization of two lone pairs at the bismuth atom. Single-crystal X-ray analyses confirmed the suggested structures of compounds **5-7** (Fig. 2b).

The bismuth atoms do not show any directional bonding interactions towards the [K(crypt)]$^+$ cations based on distance criteria. They are found in a bent coordination geometry (C−Bi−C, 92.36(18) − 97.88(16)°), with Bi−C bond lengths of 2.259(5) to 2.312(5) Å. These values are close to those reported for the corresponding dibismuthanes (e.g. Ph$_4$Bi$_2$: Bi−C, 2.26(2)/2.28(2) Å; C−Bi−C/Bi, 90.9(5) − 98.3(8)°)[49] and triaryl bismuthanes (e.g. Mes$_3$Bi: Bi−C, 2.31(1) − 2.32(1) Å; C−Bi−C, 94.7(4) − 107.6(4)°)[50]. The close similarities in Bi−C bond lengths speak against significant Bi−C multiple bond character, hinting at a considerable charge localization at the bismuth atoms of the bismuthides.

While solid samples of compounds **5-8** are stable at −30 °C for months, they slowly decompose in solution at 23 °C, resulting in the formation of bismuth black[51] and phenanthrene (for **5**), mesitylene (for **6**), benzene (for **7**) or 1,3-di*iso*propylbenzene (for **8**).

Based on the initial analyses of the simple diaryl bismuthides **5-8**, which suggested a high concentration of negative charge at the bismuth atoms, we hypothesized that inversed polarity synthetic strategies should be viable. In such approaches, the [BiR$_2$] synthon would be nucleophilic, which contrasts with established strategies for the creation of Bi$^{III}$−C bonds that typically exploit electrophilic bismuth precursors (such as bismuth halides) and reagents with nucleophilic carbon atoms (as in organo lithium compounds and Grignard reagents). This approach would promise to bring previously inaccessible structural motifs within reach. Specifically, the yet elusive bismuth acyl species R$_2$Bi−C(O)R′ were identified as particularly attractive synthetic targets. Indeed, addition of 1-adamantanecarbonyl chloride to solutions of **5** or **6** at room temperature resulted in a color change from dark orange/red to yellow, and the acyl bismuthanes **9** and **10** could be isolated in 62-84% yield as yellow (**9**) or orange (**10**) solids and fully characterized (Fig. 3a).

$^1$H NMR spectra of compounds **9** and **10** show the expected signal pattern for the aryl substituents at bismuth and three signals for the adamantyl group in the aliphatic region. Broad resonances in the $^{13}$C

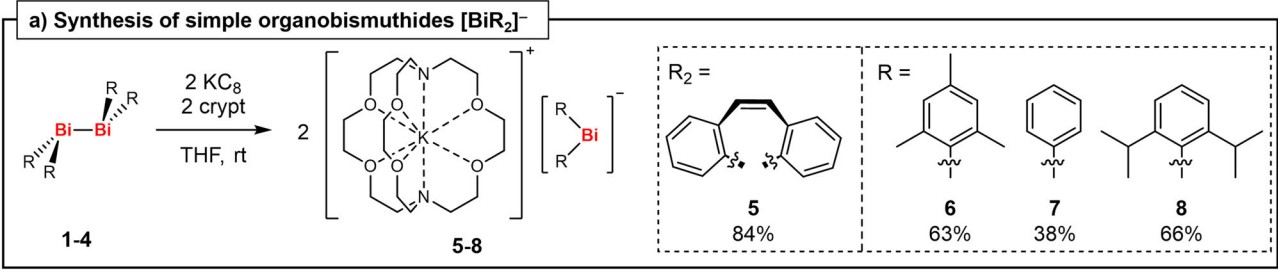

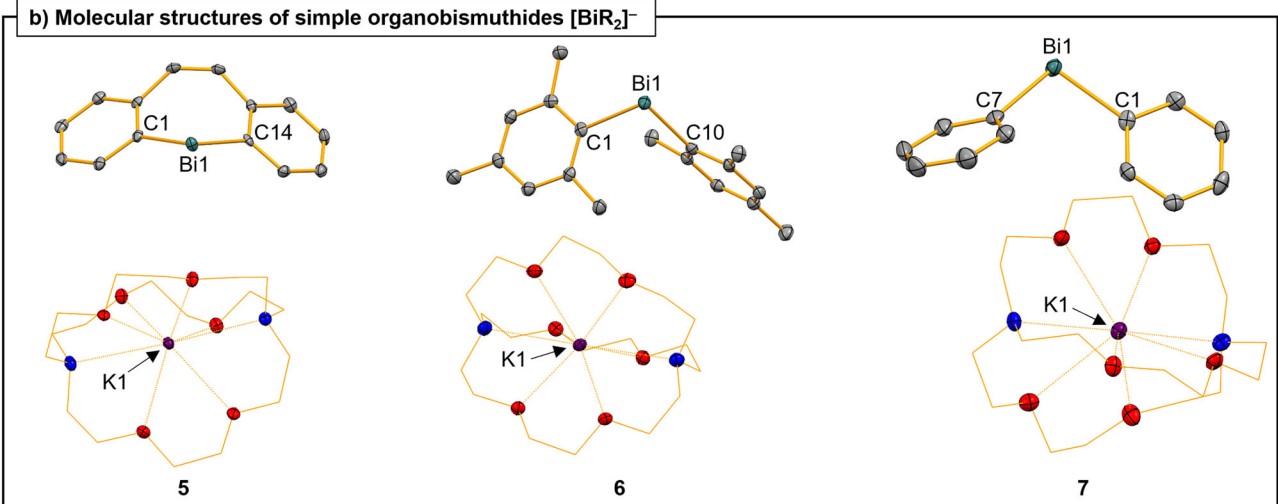

**Fig. 2 | Preparation of diaryl bismuthides and their molecular structures.**
**a** Synthesis of aryl-substituted bismuthides **5-8**; crypt = [2.2.2]cryptand. THF = tetrahydrofuran, rt = room temperature. **b** Molecular structures of compounds **5-7** in the solid state. **5** and **7** crystallize in the monoclinic space group $P2_1/c$ (Z = 4) and **6** in the triclinic space group $P\bar{1}$ (Z = 2). Displacement ellipsoids are shown at the 50% level. Hydrogen atoms are omitted and the $CH_2$ groups of [2.2.2]cryptand

displayed as wireframe for clarity. Note that XRD data of **7** (without any additional analytical data or a yield of the compound) have previously been reported with different cell parameters[41]. Selected bond lengths (Å) and angles (°): **5**, Bi1−C1/14, 2.259(5)/2.271(5), C1−Bi1−C14, 92.36(18); **6**, Bi1−C1/10, 2.307(5)/2.312(5), C1−Bi1−C10, 95.52(16); **7**, Bi1−C1/7, 2.273(4)/2.301(5), C1−Bi1−C7, 97.88(16).

NMR spectra at 253.81 (**9**) and 247.47 ppm (**10**) were assigned to the carbonyl carbon atoms. These signals show a considerable low-field shift when compared to those of lighter group 15 acyl compounds, e.g. $Me_2NC(O)Ad$ (177.0 ppm)[52] or Ter*i*PrPC(O)Ad (226.4 ppm, Ter = 2,6-bis-(2,4,6-trimethylphenyl)phenyl)[53], which is ascribed to inefficient n(Bi)→π*(C=O) interactions due to poor orbital overlap in **9** and **10**, as compared to the lighter pnictogen species. Yellow plate-shaped crystals of **9** and **10** were analyzed by single-crystal XRD (Fig. 3b). The bismuth atoms in compounds **9** and **10** show a trigonal pyramidal coordination geometry with Bi−$C_{aryl}$ bond lengths in the expected ranges. While the C−Bi−C angles in **9** are all close to 90° (88.9(2) to 92.5(2)°), the corresponding angles in **10** vary from 87.94(16) to 104.96(17)°, which was ascribed to the higher flexibility of the monodentate aryl ligands. The C=O bond lengths of 1.196(7) (**9**) and 1.211(7) Å (**10**) compare well to that in a closely related acyl phosphane (*i*Pr-TerPC(O)Ad; 1.2118(16) Å)[53]. In IR spectra of compounds **9** and **10**, the C=O stretching bands were found at wave numbers of 1691 and 1690 $cm^{-1}$, which are slightly larger than those reported for the acyl phosphane *i*PrTerPC(O)Ad (1653 $cm^{-1}$)[53] and the acyl arsane $TMS_2AsC(O)t$Bu (1667 $cm^{-1}$)[54]. Again, this is in agreement with inefficient n(Bi)→π*(C=O) interactions compared to the analogous interactions in the P and As species.

The bismuth acyl structural motif was further investigated by a quantitative Kohn-Sham molecular orbital theory analysis in combination with a quantitative energy decomposition analysis, establishing a covalent character for the hitherto unknown Bi−acyl bonding in compounds **9** and **10** (Supp. Inf., pages S44-S46). Both electrostatic and orbital interaction attractive terms equally participate to this Bi-acyl interaction (45-46% of attractive interactions), with rather small

contributions by dispersion interactions (8−9% of attractive interactions). A small charge transfer is observed from the SOMO of the [$R_2$Bi]· to the SOMO of the ·[C(O)adamantyl] fragment, which mainly involve the $p_x$(Bi) (dominant contributor) and $p_x$(C) (35%), s(C) (12%) and $p_x$(O) (31%) atomic orbitals of the CO functional group, respectively (Supp. Inf., page S46).

Experimental UV/vis spectroscopic measurements revealed broad absorption maxima at ca. 400 nm for both compounds (with a shoulder at higher wavelengths in the case of **9**), which stretch into the region of visible light. For compound **9**, TD-DFT calculations assign these bands to the HOMO-1 → LUMO and the HOMO → LUMO + 1 transitions (Fig. 3c and Supp. Inf., page S47). In particular, the aromatic backbone (HOMO-1, LUMO), the Bi−acyl group (HOMO, LUMO) and the acyl group (LUMO + 1) are involved in these transitions (for details on the analysis of **10**, see Supp. Inf., page S47).

Indeed, the light-responsiveness of compound **9** could be translated into a highly selective chemical transformation. When a solution of **9** in toluene was stored at ambient temperature and ambient light for 24 h, a color change from yellow to colorless was observed. NMR spectroscopic investigations revealed the quantitative formation (spectroscopic yield >95%) of a new bismepine species with an adamantyl substituent. Additionally, a resonance at 184.5 ppm (in $C_6D_6$) was detected in the $^{13}C$ NMR spectrum, indicating the release of carbon monoxide. Layering of the solution with *n*-pentane and storage at −30 °C led to the formation of colorless crystals, which were identified as the adamantyl-substituted bismepine **11** via single-crystal XRD (Fig. 4a,b). Compound **11**, which results from a visible-light-induced CO extrusion of the starting material, was isolated in 83% yield and fully characterized. Since the formation of **11** was strongly impeded under

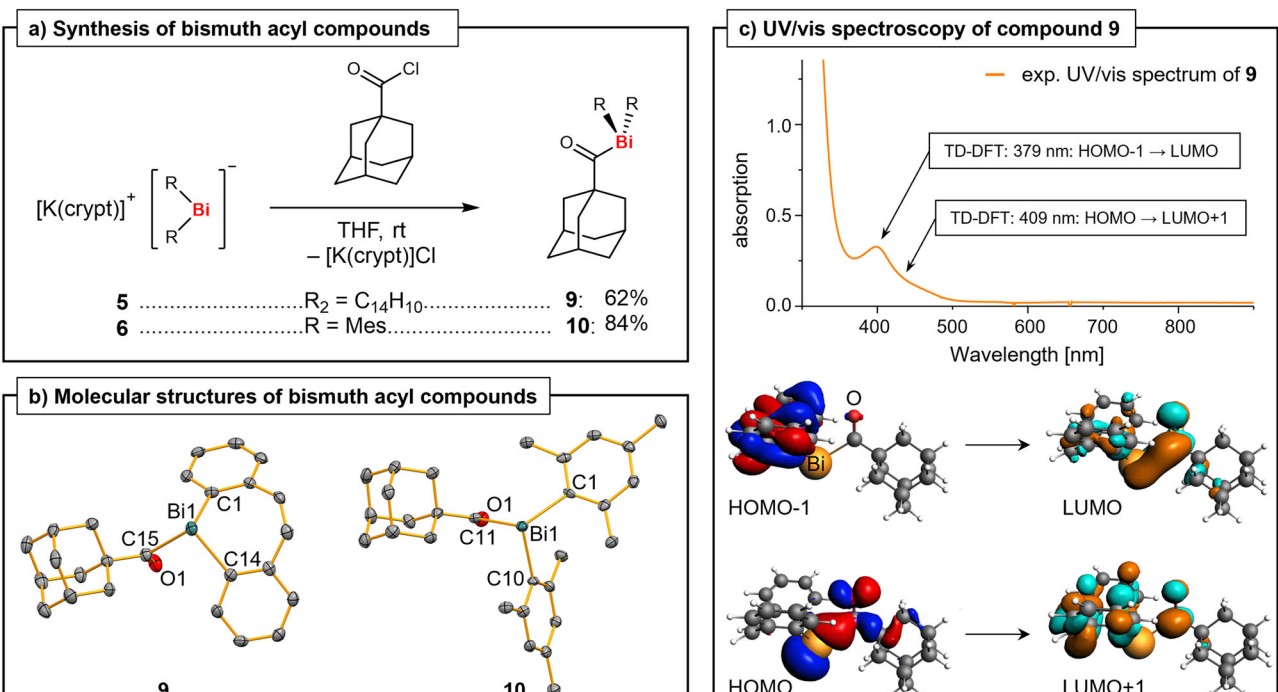

**Fig. 3 | Synthesis and characterization of acyl bismuthanes. a** Synthesis of acyl bismuthanes **9** and **10**. crypt = [2.2.2]cryptand, THF = tetrahydrofuran, rt = room temperature, Mes = mesityl = 2,4,6-trimethylphenyl. **b** Molecular structures of compounds **9** (orthorhombic space group $P2_12_12_1$, Z = 4) and **10** (orthorhombic space group $Pbca$, Z = 8) in the solid state. Displacement ellipsoids are shown at the 50% level. Hydrogen atoms are omitted for clarity. Selected bond lengths (Å) and angles (°): **9**, Bi1−C1/14, 2.233(6)/2.245(6), Bi1−C15, 2.389(5), C15−O1, 1.196(7),

C1−Bi1−C14, 92.5(2), C1/14−Bi1−C15, 88.9(2)/89.3(2); **10**, Bi1−C1/10, 2.273(4)/ 2.273(5), Bi1−C11, 2.405(5), C11−O1, 1.211(7), C1−Bi1−C10, 101.80(17), C1/10−Bi1−C11, 87.94(16)/104.96(17). **c** UV/vis spectrum of **9** in CH$_2$Cl$_2$ (0.99 mM) and results from (TD)-DFT calculations (see text; isovalue = 0.03) performed at the ZORA-BLYP-D3(BJ)/TZ2P in THF (COSMO) level of theory. UV/vis = ultraviolet-visible, exp. = experimental, TD-DFT = time-dependent density functional theory, HOMO = highest occupied molecular orbital, LUMO = lowest unoccupied molecular orbital.

exclusion of ambient light, we also investigated if the CO extrusion can be accelerated via irradiation with light of a suitable wavelength. Due to the broad absorption maximum at 400 nm in the UV/vis spectrum of **9**, a blue LED with an emission maximum at 460 nm was chosen. Irradiation of a solution of **9** in CD$_2$Cl$_2$ for 25 min led to a complete discoloration. A $^1$H NMR spectrum of the solution confirmed the formation of **11** in 81% spectroscopic yield.

Despite their similar spectroscopic properties, acyl bismuthane **10** did not show any signs of CO extrusion under ambient light, which is why more forcing photochemical conditions were examined. Irradiation of **10** in CD$_2$Cl$_2$ with a blue LED (λ = 460 nm) for 25 min led to the incomplete, but highly selective conversion into a new compound containing two mesityl units and one adamantyl group in 36% spectroscopic yield according to $^1$H NMR spectroscopy. This compound was identified as the CO extrusion product **12** through comparison of its NMR spectroscopic data with that of an independently prepared and fully characterized sample of **12** (Supp. Inf., page S9). Further irradiation under the same conditions, as well as other photochemical (LED λ = 365 nm or Hg vapor lamp) or thermal (60 °C, 50 h, in C$_6$D$_6$) conditions led to a partial and unselective decomposition of the starting material along with the formation of bismuth black[51], which was ascribed to the lack of a stabilizing chelating ligand, the steric pressure in **12** (evidenced by an extremely large C−Bi−C angle of 118°, see Supp. Inf., page S12), and the negligible involvement of the aromatic backbone in lower energy photochemical processes (Supp. Inf., page S47). Tetramesityl dibismuthane, trimesityl bismuth, and mesitylene were identified as the main products via $^1$H NMR spectroscopy.

Previous reports about reversible CO extrusion and insertion in other main group compounds[23,24,55] prompted us to investigate the reversibility of the formation of **11** and **12**. Indeed, the reaction **9** → **11** + CO was calculated to show a Gibbs free energy close to zero

(ΔG = +0.4 kcal·mol$^{-1}$ at the ZORA-BLYP-D3(BJ)/TZ2P level of theory), suggesting the possibility of an equilibrium scenario. The addition of CO (1.0 bar) to a solution of **11** in CD$_2$Cl$_2$ led to the formation of minor amounts (ca. 6%) of **9** after 4 d, according to $^1$H NMR spectroscopy indicating a partial reversibility and suggesting a thermodynamic equilibrium. Longer reaction times, higher temperatures (up to 40 °C), or Lewis basic solvents such as THF did not increase the yield considerably, while exclusion of light impeded the reaction. The addition of CO (1.0 bar) to a solution of **12** in CD$_2$Cl$_2$ resulted in an unselective and incomplete reaction. The formation of **10** (in ca. 10% spectroscopic yield) was confirmed by $^1$H and $^{13}$C{$^1$H} NMR spectroscopy, as well as mass spectrometry, proving the reversibility of the CO extrusion, albeit with lower selectivity than in the case of compounds **9/11** (see Supp. Inf., page S22).

To gain further insights into the reversibility of CO extrusion/ insertion, exchange experiments with $^{13}$C-labeled carbon monoxide were performed. $^{13}$CO (1.5 bar) was added to solutions of **9** and **10** in CD$_2$Cl$_2$, the samples were stored under ambient light at room temperature and the reactions monitored via $^1$H and $^{13}$C{$^1$H} NMR spectroscopy (for more details see Supp. Inf., pages S23-S26). For compound **9**, a sixfold increase of the intensity of the $^{13}$C NMR spectroscopic resonance assigned to its carbonyl group was observed after 25 h (Fig. 4c), despite approximately 65% of the starting material having undergone CO extrusion. While CO extrusion still occurs under an atmosphere of $^{(13)}$CO, it does so at a slower rate and without complete consumption of the starting material (approximately 16% of **9** remain, even after 16 d). These findings can be rationalized by an equilibrium between **9** and **11** + CO, where the CO extrusion is favored, even under an atmosphere of CO (as suggested in Fig. 4d). In the $^{13}$C{$^1$H} NMR spectra of the reaction of **10** and $^{13}$CO the increase of the intensity of the resonance signal of the carbonyl group was even more noticeable,

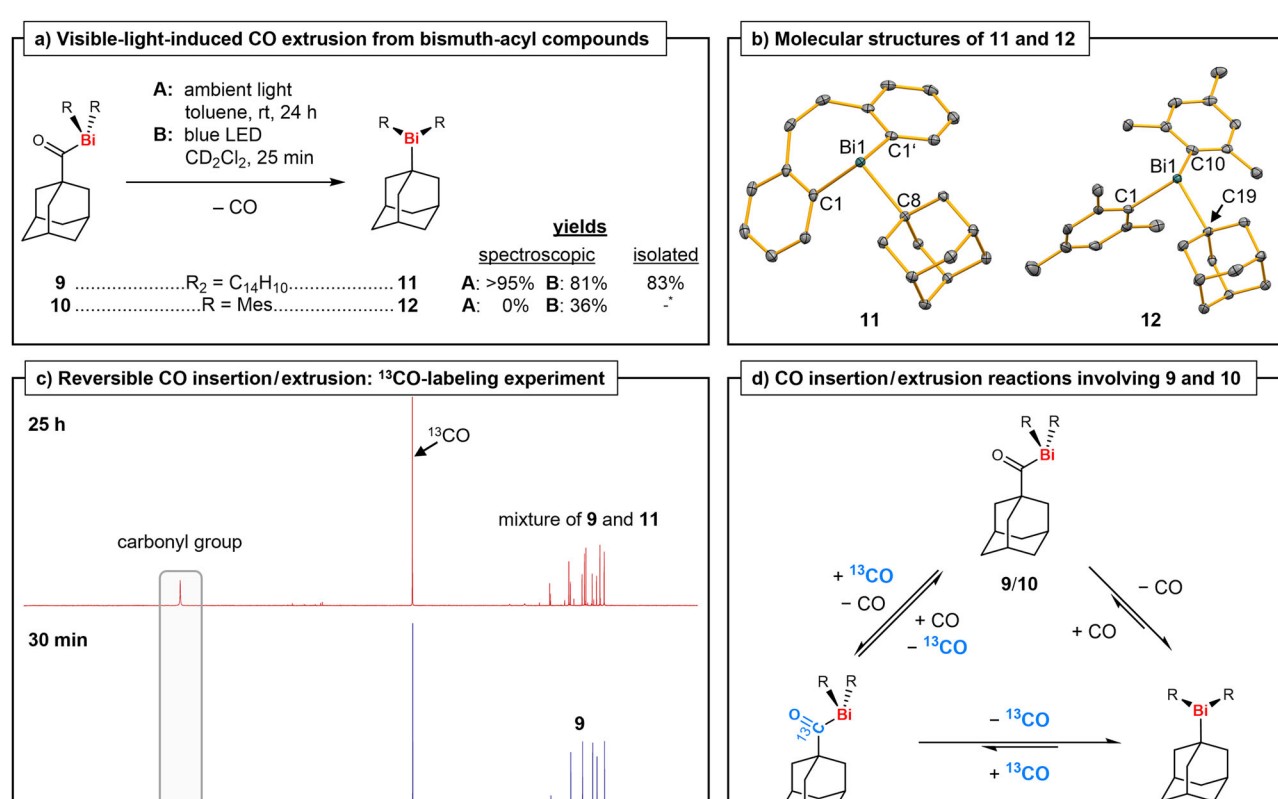

**Fig. 4 | Visible-light-induced reversible CO extrusion/insertion chemistry of acyl bismuthanes. a** Light-induced CO extrusion of **9** and **10** in solution under formation of **11** and **12**. \*: Compound **12** was prepared and isolated in 20% yield using a different synthetic approach. rt = room temperature, LED = light-emitting diode, $CD_2Cl_2$ = dichloromethane-$d_2$, Mes = mesityl = 2,4,6-trimethylphenyl. **b** Molecular structures of compound **11** (orthorhombic space group *Pnma*, Z = 4) and **12** (monoclinic space group *P2₁/n*, Z = 4) in the solid state. Displacement ellipsoids are shown at the 50% level. Hydrogen atoms are omitted for clarity.

Selected bond lengths (Å) and angles (°): **11**, Bi1 − C1, 2.2557(19); Bi1 − C8, 2.308(2); C1 − Bi1 − C1′, 90.39(10), C1 − Bi1 − C8, 100.99(6); **12**, Bi1 − C1, 2.281(4); Bi1 − C10, 2.297(3); Bi1 − C19, 2.347(3); C1 − Bi1 − C10, 97.88(13); C1 − Bi1 − C19, 92.98(12); C10 − Bi1 − C19; 118.09(13). **c** $^{13}C\{^1H\}$ NMR spectra (126 MHz) of **9** in $CD_2Cl_2$ 30 min (bottom, blue) and 25 h (top, red) after $^{13}CO$ was added. The resonance signal assigned to the carbonyl group of **9** is highlighted by a grey box. ppm = parts-per-million. **d** Equilibrium reactions rationalizing the light-induced reactivity of **9** and **10** when exposed to $^{13}CO$.

since no CO extrusion occurs under ambient light (for more details see Supp. Inf., page S27). Reversible $^{(13)}CO$ insertion/extrusion was further proved by the appearance of the resonances for the tertiary carbon atoms of the adamantyl group in [Bi]–$^{13}C(O)$adamantyl as doublets ($^1J_{CC}$ = 23 Hz (**9′**), 22 Hz (**10′**)) in the $^{13}C$ NMR spectra of these experiments.

In conclusion we have established the reliable access to simple monoanionic diaryl bismuthides, [BiR₂]⁻. Via a synthetic strategy of polarity inversion, these compounds grant access to complexes featuring the bismuth–acyl structural motif, R₂Bi−C(O)alkyl, introducing this functional group to synthetic chemistry. Exploiting the light-responsive nature of organobismuth species, this functional group paved the ground to the visible-light-driven reversible insertion and extrusion of carbon monoxide with a well-defined organometallic main group compound. While the polarity inversion approach brings previously inaccessible synthetic strategies and unknown functional groups into reach, the bismuth-acyl species promise the future exploitation of visibile-light-driven carbonyl insertion/extrusion reactions and catalysis with state-of-the art main group complexes.

## Methods
Representative methods are given below. More detailed information including all analytical data are presented in the Supplementary Information.

### Synthesis of 5
Potassium graphite (KC₈, 87 mg, 646 μmol) was added to a suspension of dibismuthane **1** (250 mg, 323 μmol) and [2.2.2]cryptand (243 mg, 646 μmol) in THF (10 mL). The reaction mixture was stirred at rt for 1 min and then filtered. The filtrate was layered with *n*-pentane (10 mL) and stored at −30 °C. After 1 d dark orange crystals of **5** had formed, which were isolated by filtration, washed with *n*-pentane (3 ×3 mL) and dried *in vacuo*. Yield: 436 mg, 543 μmol, 84%.

### Synthesis of 9
A solution of 1-adamantanecarbonyl chloride (99 mg, 498 μmol) in THF (4 mL) was added to a suspension of bismuthide **5** (400 mg, 498 μmol) in THF (10 mL). The yellow suspension was filtered and all volatiles removed under reduced pressure. The residue was extracted with a mixture of toluene (4 mL) and THF (10 mL), layered with *n*-pentane (8 mL) and stored at −30 °C. After 3 d yellow crystals of **9** had formed, which were isolated by filtration, washed with *n*-pentane (3 ×2 mL) and dried *in vacuo*. Yield: 170 mg, 309 μmol, 62%.

### $^{13}CO$ exchange experiments
In a J. Young NMR tube, **9** (30 mg, 55 μmol) was dissolved in $CD_2Cl_2$ (0.5 mL). The atmosphere above the solution was exchanged with $^{13}CO$ (1.5 bar) via three freeze/pump/thaw cycles. The reactions were monitored via $^1H$ and $^{13}C\{^1H\}$ NMR spectroscopy and the samples stored at room temperature between each measurement.

## Data availability

The data that support the findings of this study (NMR spectra, UV/vis spectra, mass spectrometric data, cartesian coordinates from DFT calculations) are available within the main text and its Supplementary Information. Crystallographic data for the structures reported in this Article have been deposited at the Cambridge Crystallographic Data Centre, under deposition numbers CCDC 2447486 (**5**), 2447487 (**6**), 2447488 (**7**), 2447489 (**9**), 2447490 (**10**), 2447491 (**11**), and 2488909 (**12**). Copies of the data can be obtained free of charge via https://www.ccdc.cam.ac.uk/structures/. All data are available from the corresponding author upon request. Source data are provided with this paper.

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

## Acknowledgements

Funding by the Deutsche Forschungsgemeinschaft (DFG, grant number LI2860/5-1; C. L.), the LOEWE program (LOEWE/4b//519/05/ 01.002(0002)/85; C. L.), and the Spanish Ministerio de Ciencia, Innovación y Universidades (MCIN/AEI/10.13039/501100011033; J. P.) for projects PID2022-138861NB-I00 (J. P.) and CEX2021-001202-M (J. P.). This project has received funding from the European Research Council (ERC) under the European Union's Horizon 2020 research and innovation program (grant agreement No 946184 C. L.). The authors thank Dr. Jacqueline Ramler and Leonie Wüst for preliminary work towards the synthesis of diaryl bismuthides.

## Author contributions

Conceptualization: F.G. and C.L.; Formal Analysis (Theoretical Chemistry Part): J.P.; Funding Acquisition, J.P. and C.L.; Project Administration and Resources: C.L.; Investigation: F.G., B.N., D.B., T.D., S.R.; Visualization: F.G., J.P., C.L.; Supervision: F.G. (minor) and C.L. (major); Writing—original draft: F.G., J.P., and C.L.; Writing—review and editing: All authors.

## Funding

## Competing interests

The authors declare no competing interests.
