## [Peer Review File · Nature Communications]

Diaryl Bismuthides and Acyl Bismuthanes Enable Visible-Light-Induced Reversible Carbon Monoxide Insertion and Extrusion

Corresponding Author: Professor Crispin Lichtenberg

Version 0:

Reviewer comments:

Reviewer #1

(Remarks to the Author)

This manuscript describes the synthesis and spectroscopic characterization of four diorganobismuthide (R_2Bi-) compounds (5-8), as well as the X-ray crystal structures of three of these compounds (5-7). Compounds 5 and 6 were employed as nucleophilic synthons in reactions with 1-adamantanecarbonyl chloride to afford the corresponding bismuth acyl species $R_2Bi-C(O)R'$ (9 and 10), both of which were spectroscopically and structurally characterized. The nature of the Bi-C(O) bond was examined computationally and the results suggest more covalent versus ionic character. Further, absorptions for both compounds at ~ 400 nm were observed for 9 and 10 in the UV-visible absorption spectrum and transitions were assigned with the aid of TD-DFT calculations. Observations of a color change in toluene solutions of compound 9 stored in ambient light prompted the authors to further study the phenomenon. NMR spectroscopy confirmed loss (extrusion) of CO from compound 9 and formation of the corresponding triorganobismuthine R_2Bi-R' (11), the latter of which was isolated and fully characterized. The authors also demonstrate CO extrusion in compound 9 can be accelerated by blue light (460 nm) irradiation, while compound 11 can be converted back to the bismuth acyl species (9) in low yield in the presence of a CO atmosphere. Finally, solutions of bismuth acyl compound 9 held under an atmosphere of ^{13}C -labelled carbon monoxide show exchange of CO, indicating a reversible CO insertion/extrusion and an equilibrium between 9 and 11 + CO.

Overall, the manuscript is very well written. The reported compounds are sufficiently characterized, all experiments and computational studies are well thought out and adequately described, and pertinent literature has been referenced. The diorganobismuthide compounds are useful synthons for the formation of novel bismuth compounds, as demonstrated in the synthesis of previously unreported acyl bismuth compounds (9 and 10). The most significant result is the observations of reversible insertion/extrusion of CO into a bismuth-carbon bond, a phenomenon not observed for main group elements and an important step in the development of main group element-based catalysts. The findings are novel and significant and the manuscript merits publication in Nature Communication after the following points are addressed:

Major

1. The authors highlight the significance of the reversible insertion/extrusion of CO into the bismuth-carbon bond and suggest this as a large step forward in establishing main group element catalysis. However, this was achieved for compound 9 (11) only. Attempts to induce the same reactivity in compound 10 were unsuccessful and resulted in decomposition of the compound (lines 178-183 in the manuscript). Further, diorganobismuthides 7 and 8 were not converted to the corresponding bismuth acyl species, and were not tested for ability to achieve reversible insertion/extrusion of CO. The authors rationalize the decomposition of 10 versus 9 as "a lack of stabilizing chelate ligand and negligible involvement of the aromatic backbone in the lower energy chemical process." Given that these are only two data points and the conclusion is speculative, it would greatly improve the impact of the manuscript and general applicability of its findings if the authors could demonstrate the chemical reversibility in a second bismuth-acyl compound. This could involve another chelating ligand, or perhaps a substituted version of the chelating ligand employed in compound 9. Use of the latter may result in increased reactivity depending on the choice of substituents. This would also help establish this study as an important step in establishing "detailed understanding and control of these elementary reactions in the chemistry of carbon monoxide with main group elements [as] the basis for the development of selective and competitive transformation." (lines 44-46)

Minor

2. line 106 – add a range of C-Bi-C and Bi-Bi-C bond angles for Ph₄Bi₂ and Mes₃Bi for comparison to those in compounds 5-7.
3. lines 123-127 – Comment on why the carbonyl ¹³C signals show significant low field shift verses those of lighter group 15 acyl compounds.
4. lines 133-135 – Comment on why the C=O stretching frequencies of 9 and 10 are higher than those of acyl phosphane and arsane compounds.

Reviewer #2

(Remarks to the Author)

The work by Lichtenberg's group describes the reversible carbon monoxide extrusion/insertion at a Bi(III) center in bismuthanes. This study represents a valuable addition to the growing field of CO activation by main group elements, which can mimic the behavior of transition metals. While CO activation by main group species has been demonstrated in low-valent compounds, multiple-bonded species, geometrically constrained systems, and frustrated Lewis pairs, a detailed understanding of the elementary steps and model systems that exhibit reversible CO transformations remains fundamentally important for advancing selective and competitive CO-based reactivity.

In this context, reversible CO insertion/extrusion has only been reported in a limited number of cases involving boron and aluminum, with one photochemical example for germanium. The present discovery of reversible CO extrusion/insertion at a Bi center in bismuth acyl compounds significantly expands the scope of this chemistry. The novelty and conceptual advance presented in this work are suitable for publication in Nature Communications.

The experimental work is thorough, with careful characterization of the synthesized compounds by NMR, IR, UV-Vis, HRMS, single-crystal XRD, and elemental analysis. The electronic properties of key compounds are also well investigated by DFT calculations. The central conclusion—reversible CO extrusion/insertion—is strongly supported by experiments, including those using ¹³C-labeled CO. While the light-induced CO extrusion is clearly demonstrated, there remains some ambiguity regarding the role of light in the insertion process.

Major Comments

--Since the reversible CO insertion/extrusion is only observed for 9 featuring a cyclic bismepine moiety, while 10 bearing more general aryl groups does not exhibit such reactivity, the depiction in the right panel of Figure 1c should be revised accordingly to accurately reflect this limitation.

--The reaction of 11 with CO under ambient light at room temperature is extremely slow, yielding only ~5% of acyl bismuthane 9 after 28 days. Does this suggest that the formation of 9 is thermodynamically unfavorable or that the kinetic barrier is prohibitively high? A discussion in the main text would be appropriate. Additionally, did the authors attempt this reaction under elevated temperatures or with LED light irradiation to accelerate the conversion? Is the CO insertion thermally promoted?

--Although the light-driven CO extrusion from 9 to 11 is well supported by experiments, additional control studies are necessary to confirm whether light is also required for the reverse CO insertion (from 11 to 9). This key aspect is not discussed in the main text. The authors are encouraged to include further control experiments, possibly in the Supporting Information, to strengthen their mechanistic claims.

Minor Comments

--Exploring potential reactivity of 9 would substantially enhance the impact of this work. For example, has the reactivity of 9 toward small molecules such as H₂ been investigated?

--Page 2, line 62: Please be cautious with the phrase "the first set of isolable diaryl bismuthides." Although prior literature may not include fully characterized examples, the generation and transformation of diaryl bismuthides as reactive intermediates, and the crystal structure of [K(crypt)][BiPh₂], have been reported (Refs 40–41). While the authors present a new synthetic route and thorough characterization, the claim of "first" is overstated.

--Page 2, line 63: The term "unprecedented inverse-polarity approach" should be reconsidered. Inverse-polarity reactions involving diaryl bismuthides and electrophiles have already been documented (Refs 38, 40). In this work, the electrophile is an acyl chloride. Thus, the use of "unprecedented" appears exaggerated.

--Page 3, line 78: The term "elusive" to describe bismuthides is misleading and should be revised.

--Page 3, line 79: The statement "But the rational synthesis, isolation, and characterization of these compounds have not been reported to date" should be modified. Previous work by Suzuki and others (Ref 40) should not be overlooked.

--Page 3, line 75: The phrase "without stabilizing electronic effects" is problematic. Any scaffold or substituent inherently exerts some electronic influence. Please revise this statement for accuracy.

--Figure 4d: Consider removing the equilibrium arrow between 9 and 9' to better illustrate the transformation pathway

between 9, 11, and 9'.

--Could kinetic data—such as apparent rate constants, activation energy, and activation parameters (apparent rate constants, activation energy, enthalpy and entropy of activation)—be extracted from the ^{13}C O labeling experiments involving 9? Such information would provide a deeper understanding of the reaction mechanism.

--In the Supporting Information, the mention of compound 3 seems out of place and possibly erroneous:

--Table S5: Data obtained from ^1H and $^{13}\text{C}\{^1\text{H}\}$ NMR spectra of reaction of 3 with ^{13}C O in CD_2Cl_2 .

--When the same reaction was performed and the sample stored under exclusion of light between measurements (as far as possible), no signs for CO extrusion or the incorporation of ^{13}C O into 3 were observed, giving further evidence of the light-driven nature of the reaction.

--Please clarify whether this section refers to 3 correctly or if it should refer to 9.

--Figure S17: The peak at 146.1 ppm is not visible. Please zoom in on this region of the ^{13}C NMR spectrum for clarity.

Reviewer #3

(Remarks to the Author)

Version 1:

Reviewer comments:

Reviewer #1

(Remarks to the Author)

My comments have all been addressed in the revised manuscript. I commend the authors on carrying out the additional experiment required to demonstrate the reversible CO insertion/extrusion in compounds 10/12. This greatly enhances impact of the manuscript, supports the general applicability of its findings, and furthers the development of main group element-based catalysts. I recommend publication of the manuscript in Nature Communications in its current form.

Reviewer #2

(Remarks to the Author)

The authors have revised their manuscript and uploaded a suitable manuscript. I am thus happy to recommend publication of this work in Nat. Commun. without changes. Congrats!

Reviewer #3

(Remarks to the Author)

Point-by-point response to the reviewers' comments

REVIEWER COMMENTS

Reviewer #1 (Remarks to the Author):

This manuscript describes the synthesis and spectroscopic characterization of four diorganobismuthide (R_2Bi-) compounds (5-8), as well as the X-ray crystal structures of three of these compounds (5-7). Compounds 5 and 6 were employed as nucleophilic synthons in reactions with 1-adamantanecarbonyl chloride to afford the corresponding bismuth acyl species $R_2Bi-C(O)R'$ (9 and 10), both of which were spectroscopically and structurally characterized. The nature of the $Bi-C(O)$ bond was examined computationally and the results suggest more covalent versus ionic character. Further, absorptions for both compounds at ~ 400 nm were observed for 9 and 10 in the UV-visible absorption spectrum and transitions were assigned with the aid of TD-DFT calculations. Observations of a color change in toluene solutions of compound 9 stored in ambient light prompted the authors to further study the phenomenon. NMR spectroscopy confirmed loss (extrusion) of CO from compound 9 and formation of the corresponding triorganobismuthine R_2Bi-R' (11), the latter of which was isolated and fully characterized. The authors also demonstrate CO extrusion in compound 9 can be accelerated by blue light (460 nm) irradiation, while compound 11 can be converted back to the bismuth acyl species (9) in low yield in the presence of a CO atmosphere. Finally, solutions of bismuth acyl compound 9 held under an atmosphere of ^{13}C -labelled carbon monoxide show exchange of CO, indicating a reversible CO insertion/extrusion and an equilibrium between 9 and 11 + CO.

Overall, the manuscript is very well written. The reported compounds are sufficiently characterized, all experiments and computational studies are well thought out and adequately described, and pertinent literature has been referenced. The diorganobismuthide compounds are useful synthons for the formation of novel bismuth compounds, as demonstrated in the synthesis of previously unreported acyl bismuth compounds (9 and 10). The most significant result is the observations of reversible insertion/extrusion of CO into a bismuth-carbon bond, a phenomenon not observed for main group elements and an important step in the development of main group element-based catalysts.

The findings are novel and significant and the manuscript merits publication in Nature Communication after the following points are addressed:

REPLY: We thank reviewer 1 for the positive evaluation of our work and the time invested to provide his/her comments.

Major

1. The authors highlight the significance of the reversible insertion/extrusion of CO into the bismuth-carbon bond and suggest this as a large step forward in establishing main group element catalysis. However, this was achieved for compound 9 (11) only. Attempts to induce the same reactivity in compound 10 were unsuccessful and resulted in decomposition of the compound (lines 178-183 in the manuscript). Further, diorganobismuthides 7 and 8 were not converted to the corresponding bismuth acyl species, and were not tested for ability to achieve reversible insertion/extrusion of CO. The authors rationalize the decomposition of 10 versus 9 as "a lack of stabilizing chelate ligand and negligible involvement of the aromatic backbone in the lower energy chemical process." Given that these are only two data points and the conclusion is speculative, it would greatly improve the impact of the manuscript and general applicability of its findings if the authors could demonstrate the chemical reversibility in a second bismuth-acyl compound. This could involve another chelating ligand, or perhaps a substituted

version of the chelating ligand employed in compound 9. Use of the latter may result in increased reactivity depending on the choice of substituents.

This would also help establish this study as an important step in establishing “detailed understanding and control of these elementary reactions in the chemistry of carbon monoxide with main group elements [as] the basis for the development of selective and competitive transformation.” (lines 44-46)

REPLY:

We thank reviewer 1 for these valuable suggestions. We have performed a range of experiments inspired by these helpful comments (supported by a detailed literature survey).

Bismuthides with chelating aryl ligands. We agree that another set of compounds with a chelating ligand coordinated to the bismuth center would potentially be very helpful to further understand the differences in reactivity of compounds **9** and **10**. An exhaustive literature survey shows that there are only three examples in the literature which describe the synthesis of dibismuthanes with chelating aryl ligands. This includes our own work on the synthesis of dibismuthane **1**, which is exploited in the current manuscript. The remaining two types of dibismuthanes with chelating aryl ligands are i) compounds featuring a cyclic azabismocine framework (*Chem. Commun.* **2009**, 6168) and ii) a compound with a thioether ligand backbone prepared by our group (*Chem. Eur. J.* **2020**, *26*, 14551). Thus, the field of dibismuthanes with chelating aryl ligands is poorly explored to date. When it comes to the utilization of the literature-known compounds as precursors for bismuthide chemistry, two problems come up. Firstly, these species bear benzylic positions with a heteroatom (N or S) in α -position, which is typically prone to C-H or C-N/S activation under strongly basic or strongly reducing conditions (as necessary for the synthesis of bismuthides). In addition, both types of compounds are poorly soluble, even in polar media such as THF or pyridine (for instance: for the extraction of 59 mg of one of the products 100 mL of THF had to be used; ^{13}C NMR data have not been reported in the case of the azabismocines and have only been obtained in hot pyridine (70 °C) relying on the higher sensitivity of 2D NMR spectroscopy in the other case).

Modifications of the dibenzobismepine framework have been reported to a limited extent, but not to the stage of precursors for dibismuthane synthesis. More importantly, these species lack the stabilizing effect of showing two Bi-C^{Aryl} bonds and can be extremely labile (e.g.: half life times of 7 minutes (60 °C, toluene) have been reported (*J. Chem. Soc., Chem. Commun.* **1993**, 1817)).

Thus, we set out to synthesize other types of dibismuthanes with chelating aryl ligands, which avoid the problem of reactive benzylic positions. Our efforts indeed led to the successful synthesis of a new dibismuthane [Bi(C₆H₄)₂SO₂]₂. This compound could be structurally characterized by XRD analysis of single crystals, which had been grown from the reaction medium. However, this compound is very poorly soluble in common organic solvents (benzene, dichloromethane, THF, pyridine, acetonitrile). The solubility issues are pronounced to an extent that not even a meaningful ^1H NMR spectrum could be obtained, thereby hampering further selective transformations towards the selective synthesis of bismuthides.

We conclude that the synthesis of dibismuthanes with properties to allow for the subsequent preparation of electronically non-stabilized bismuthides (a new class of compounds presented in this manuscript) is an important goal, but will require synthetic efforts that exceed the scope of the current work.

Detailed re-investigation of BiMes₂(COAd) (10). Inspired by this reviewer’s comments, we thus decided to carefully re-investigate if more detailed information can be obtained on the degradation of BiMes₂(COAd) (**10**) that had been reported in the original version of the manuscript. We performed a close reaction monitoring of irradiation experiments with compound **10** under various conditions,

using small increments in the time dimension. These investigations confirmed that (as originally reported) unselective degradation reactions take place when **10** is subjected to long periods of irradiation. However, our efforts also revealed that indeed, the selective formation of a new species can be observed when **10** is irradiated for 25 minutes with a blue LED ($\lambda = 460$ nm) in 36% spectroscopic yield (with 64% of the starting material remaining unreacted). We hypothesized that the new compound BiMes₂Ad (**12**) is generated under these conditions upon release of carbon monoxide. In order to unambiguously confirm the selective CO extrusion, we aimed to independently prepare compound **12**. Severe synthetic challenges of ligand redistribution (typical for bismuth compounds of the type BiR₂R', e.g.: *Organometallics* **2020**, *39*, 778) and the selective generation of sufficiently nucleophilic adamantyl species [M]-Ad in the absence of other reactive compounds (e.g. excess of initial reductants or transmetalating agents) (*J. Org. Chem.* **1983**, *48*, 2975, *J. Am. Chem. Soc.* **1982**, *104*, 3481, *Organometallics* **1989**, *8*, 2664, *Org. Synth.* **1995**, *72*, 147) required a tedious screening of approaches and reaction conditions. A successful protocol could be established by exploiting the organo-catalyzed in-situ generation of Rieke calcium, followed by the in-situ generation of the “heavy Grignard analog” CaAdBr, followed by a salt elimination to give the desired bismuth compound. Despite this multistep protocol involving various highly reactive species, compound **12** could be isolated and fully characterized (revealing extreme bond angles due to steric pressure, rationalizing its high reactivity towards CO (*vide infra*)). Comparison with authentic sample confirmed that indeed, compound **10** undergoes selective CO extrusion to give **12** in 36% spectroscopic yield, paralleling the observations we have made with the bismepine species **9** and **11**. The isolation of **12** also allowed us to investigate its reactivity towards (¹³C-labeled) carbon monoxide. These experiments confirmed that the insertion of CO into the Bi–C^{Ad} bond of **12** with formation of the acyl bismuthane **10** is possible. In addition, exposure of BiMes₂(COAd) (**10**) to ¹³CO gave the ¹³C-labeled compound BiMes₂(¹³COAd) (**10'**). Altogether, these new results confirm that the reversible photochemically driven extrusion/insertion of CO from/with well-defined organobismuth compounds is possible. It is not only observed with compound **9** (supported by a chelating ligand), but also for the simpler compound **10** (albeit with lower selectivity).

We would like to explicitly thank reviewer 1 for his/her valuable comments that led to the tedious but rewarding in-depth analysis of the CO extrusion/insertion behavior of compounds **10/12** under photochemical conditions.

The new insights have been incorporated into the revised version of the manuscript and the supplementary information as follows:

Manuscript:

“Despite their similar spectroscopical properties acyl bismuthane **10** did not show any signs of CO extrusion under ambient light, which is why more forcing photochemical conditions were examined. Irradiation of **10** in CD₂Cl₂ with a blue LED ($\lambda = 460$ nm) for 25 min led to the incomplete, but highly selective conversion into a new compound containing two mesityl units and one adamantyl group in 36% spectroscopic yield according to ¹H NMR spectroscopy. This compound was identified as the CO extrusion product **12** through comparison of its NMR spectroscopic data with that of an independently prepared and fully characterized sample of **12** (Supp. Inf.). Further irradiation under the same conditions, as well as other photochemical (LED $\lambda = 365$ nm or Hg vapor lamp) or thermal (60 °C, 50 h, in C₆D₆) conditions led to a partial and unselective decomposition of the starting material along with the formation of “bismuth black”, which was ascribed to the lack of a stabilizing chelating ligand, the steric pressure in **12** (evidenced by an extremely large C–Bi–C angle of 118°, see Supp. Inf.), and the

negligible involvement of the aromatic backbone in lower energy photochemical processes. Tetramesityl dibismuthane, trimesityl bismuth, and mesitylene were identified as the main products via ^1H NMR spectroscopy.

Figure 4. a) Light-induced CO extrusion of **9** and **10** in solution under formation of **11** and **12**. *: Compound **12** was prepared and isolated in 20% yield using a different synthetic approach. b) Molecular structures of compound **11** (orthorhombic space group $Pnma$, $Z = 4$) and **12** (monoclinic space group $P2_1/n$, $Z = 4$) in the solid state. Displacement ellipsoids are shown at the 50% level. Hydrogen atoms are omitted for clarity. Selected bond lengths (\AA) and angles ($^\circ$): **11**, Bi1–C1, 2.2557(19); Bi1–C8, 2.308(2); C1–Bi1–C1', 90.39(10), C1–Bi1–C8, 100.99(6); **12**, Bi1–C1, 2.281(4); Bi1–C10, 2.297(3); Bi1–C19, 2.347(3); C1–Bi1–C10, 97.88(13); C1–Bi1–C19, 92.98(12); C10–Bi1–C19; 118.09(13). c) $^{13}\text{C}\{^1\text{H}\}$ NMR spectra of **9** in CD_2Cl_2 30 min (bottom, blue) and 25 h (top, red) after ^{13}CO was added. The resonance signal assigned to the carbonyl group of **9** is highlighted by a grey box. d) Equilibrium reactions rationalizing the light-induced reactivity of **9** and **10** when exposed to ^{13}CO ."

"The addition of CO (1.0 bar) to a solution of **12** in CD_2Cl_2 resulted in an unselective and incomplete reaction. The formation of **10** (in ca. 10% spectroscopic yield) was confirmed by ^1H and $^{13}\text{C}\{^1\text{H}\}$ NMR spectroscopy, as well as mass spectrometry, proving the reversibility of the CO extrusion, albeit with lower selectivity than in the case of compounds **9/11** (see Supp. Inf.)."

"To gain further insights into the reversibility of CO extrusion/insertion, exchange experiments with ^{13}C -labeled carbon monoxide were performed. ^{13}CO (1.5 atm) was added to solutions of **9** and **10** in CD_2Cl_2 , the samples were stored under ambient light at room temperature and the reactions monitored via ^1H and $^{13}\text{C}\{^1\text{H}\}$ NMR spectroscopy (for more details see Supp. Inf.). For compound **9**, a

sixfold increase of the intensity of the ^{13}C NMR spectroscopic resonance assigned to its carbonyl group was observed after 25 h (Figure 4c), despite approximately 65% of the starting material having undergone CO extrusion. While CO extrusion still occurs under an atmosphere of ^{13}C CO, it does so at a slower rate and without complete consumption of the starting material (approximately 16% of **9** remain, even after 16 d). These findings might be rationalized by an equilibrium between **9** and **11** + CO, where the CO extrusion is favored, even under an atmosphere of CO (as suggested in Figure 4d). In the $^{13}\text{C}\{^1\text{H}\}$ NMR spectra of the reaction of **10** and ^{13}C CO the increase of the intensity of the resonance signal of the carbonyl group was even more noticeable, since no CO extrusion occurs under ambient light (for more details see Supp. Inf.). Reversible ^{13}C CO insertion/extrusion was further proved by the appearance of the resonances for the tertiary carbon atoms of the adamantyl group in $[\text{Bi}]-^{13}\text{C}(\text{O})\text{adamantyl}$ as a doublet ($^1J_{\text{CC}} = 23 \text{ Hz}$ (**9'**), 22 Hz (**10'**)) in the ^{13}C NMR spectra of these experiments.”

Supplementary Information:

“Synthesis of Mes_2BiAd (**12**)

The synthesis was based on a literature procedure for the preparation of 1-adamantyl calcium halides.⁶ Elemental lithium (16 mg, 2.32 mmol) and biphenyl (359 mg, 2.32 mmol) were stirred in THF (10 mL) at rt for 3 h. The dark-green solution was added to a suspension of CaI_2 (341 mg, 1.16 mmol), and the reaction mixture was stirred for 1 h at rt. The dark-orange suspension was cooled to $-78 \text{ }^\circ\text{C}$ and a solution of 1-bromoadamantane (250 mg, 1.16 mmol) in THF (7 mL) was added dropwise. After stirring for 20 min at $-78 \text{ }^\circ\text{C}$ a suspension of dimesityl bismuth chloride (432 mg, 895 μmol) in a mixture of THF (10 mL) and toluene (4 mL) was added dropwise. The suspension was warmed to room temperature overnight, resulting in the formation of a black suspension. All volatiles were removed under reduced pressure, and the residue was extracted with *n*-pentane (10 mL) and *n*-hexane (4 mL). The solvent mixture was allowed to evaporate at $-30 \text{ }^\circ\text{C}$, which resulted in two batches of solid material after one and two days respectively. The solids were isolated by filtration and dried *in vacuo*. The first batch (151 mg, off-white powder) was identified as biphenyl via ^1H NMR spectroscopy. The second batch consisted of analytically pure, pale-orange crystals of **12**. **Yield:** 102 mg, 175 μmol , 20%.

^1H NMR (500 MHz, CD_2Cl_2): $\delta = 1.75\text{--}1.94$ (m, 6H, Ad- CH_2), $2.03\text{--}2.08$ (m, 3H, Ad-CH), 2.21 (s, 6H, CH_3 -*para*), 2.23 (s, 12H, CH_3 -*ortho*), $2.41\text{--}2.44$ (m, 6H, Ad- CH_2), 6.94 (s, 4H, Ar-*H*) ppm. $^{13}\text{C}\{^1\text{H}\}$ NMR (126 MHz, CD_2Cl_2): $\delta = 21.12$ (s, CH_3 -*para*), 28.87 (s, CH_3 -*ortho*), 34.12 (s, Ad-CH), 37.26 (s, Ad- CH_2), 45.66 (s, Ad- CH_2), 60.48 (s, Ad-C) 129.25 (s, ArC-*meta*), 137.00 (s, ArC-*para*), 145.33 (s, ArC-*ortho*), 161.65 (s, ArC-*ipso*) ppm. **HR-MS (APCI, pos.):** calculated for $(^{12}\text{C}_{18}\text{H}_{22}^{209}\text{Bi})^+$ (Mes_2Bi^+): $m/z = 447.1520$, found: $m/z = 447.1507$, calculated for $(^{12}\text{C}_{19}\text{H}_{26}^{209}\text{Bi})^+$ (MesAdBi^+): $m/z = 463.1833$, found: $m/z = 463.1821$. The molecular ion peak was not detected in multiple experiments using the soft ionization methods APCI or LIFDI. **Elemental analysis:** Anal. calc. (%) for $\text{C}_{28}\text{H}_{37}\text{Bi}$ ($582.58 \text{ g} \cdot \text{mol}^{-1}$): C 57.73, H 6.40; found: C 58.07, H 6.05.”

“Adamant-1-yl-dimesitylbismuthane **12** crystallizes in the monoclinic space group $P2_1/n$ with $Z = 4$ (Figure S1). The molecular structure shows the bismuth atom in a distorted trigonal-pyramidal coordination geometry (C–Bi–C, $92.98(12)$ – $118.09(13)^\circ$), with Bi–C_{aryl} bond lengths (2.281(4)/2.297(3) Å) similar to those in the mesityl-substituted acylbismuthane **10** (2.273(4)/2.273(5) Å). The Bi–C_{Ad} bond length of 2.347(3) Å is slightly longer than that in the closely related bismepine species **11** (2.308(2) Å), most likely due to the higher steric demand of the mesityl substituents. To the best of our knowledge, the C10–Bi1–C19 angle ($118.09(13)^\circ$) is the highest reported so far for trivalent bismuthanes bearing three carbon-based substituents. A higher C–Bi–C angle has only been reported for a diarylbismuth chloride bearing two bulky terphenyl substituents ((2,6-Mes₂H₃C₆)₂BiCl, $123.9(3)^\circ$).⁸

Figure S1. Molecular structure of compound **12** in the solid state (monoclinic space group $P2_1/n$, $Z = 4$). Displacement ellipsoids are shown at the 50% level. Hydrogen atoms are omitted for clarity. Selected bond lengths (Å) and angles ($^\circ$): Bi1–C1, 2.281(4); Bi1–C10, 2.297(3); Bi1–C19, 2.347(3); C1–Bi1–C10, 97.88(13); C1–Bi1–C19, 92.98(12); C10–Bi1–C19; 118.09(13).”

“Reactivity of **12** towards CO

In a J. Young NMR tube, **12** (12 mg, 21 μ mol) was dissolved in CD₂Cl₂ (0.5 mL). The atmosphere above the solution was exchanged with CO (1.0 bar) *via* three freeze/pump/thaw cycles. The reaction was monitored via ¹H NMR spectroscopy and the sample stored at room temperature under ambient light between each measurement. A ¹H NMR spectrum of the reaction mixture after 1 h showed the resonances of the starting material and multiple new resonance signals in the aromatic, as well as the aliphatic region, indicating an incomplete and unselective reaction (>10 products). While a longer reaction time (up to 48 h) led to a slight decrease of the intensity of the resonance signals assigned to **12**, the reaction remained incomplete, and the product distribution did not change significantly. The acyl bismuthane **10** was identified as one of the products by comparison of the ¹H and ¹³C{¹H} NMR data obtained from the reaction with that of a pure sample.

Figure S12: Aromatic region of the ^1H NMR spectrum of the reaction of **12** with CO in CD_2Cl_2 after 48 h.

10 is formed in approximately 10% spectroscopic yield (value after 48 h, but only minor changes to this are observed at earlier stages of the reaction) and its formation was additionally confirmed by the detection of its molecular ion peak in a high-resolution mass spectrum of the reaction mixture (LIFDI, positive mode, Figure S13).

Figure S13: Experimental LIFDI-MS spectrum of the reaction mixture after 48 h (positive mode, top) and simulated isotope pattern of the cation $[\mathbf{10}]^+$ (bottom)."

"Reactivity of **10** towards ^{13}C O

In a J. Young NMR tube, **10** (15 mg, 25 μmol) was dissolved in CD_2Cl_2 (0.5 mL). The atmosphere above the solution was exchanged with ^{13}C O (1.5 bar) *via* three freeze/pump/thaw cycles. The reaction was monitored via ^1H and $^{13}\text{C}\{^1\text{H}\}$ NMR spectroscopy and the sample stored at room temperature under ambient light between each measurement. To gain insights into the incorporation of ^{13}C O into **10**, the

relative intensity of the $^{13}\text{C}\{^1\text{H}\}$ NMR resonance signal assigned to the carbonyl C atom of **10** was determined using the solvent signal as a reference (integral set to a value of 1.00, Table S6).

Table S6: Data obtained from ^1H and $^{13}\text{C}\{^1\text{H}\}$ NMR spectra of the reaction of **10** with ^{13}CO in CD_2Cl_2 .

	before	30 min	24 h	7 d	9 d	10 d
^{13}C NMR integral carbonyl group of 10	0.015	0.050	0.318	0.463	0.466	0.519

After 24 h, a 21-fold increase of the intensity of the ^{13}C NMR spectroscopic resonance assigned to the carbonyl group of **10** ($\delta = 247.47$ ppm) was observed, indicating that ^{13}CO is incorporated into compound **10** and proving the reversibility of the CO extrusion. Over the course of 10 d the intensity of the resonance signal continued to increase up to a factor of about 35 (when compared to a $^{13}\text{C}\{^1\text{H}\}$ NMR spectrum of the same sample before the addition of ^{13}CO), after which no further increase was observed. During the reaction, the slow formation of small amounts of mesitylene, BiMe_3 and other unidentified products was observed spectroscopically with about 73% of **10** being detected after 10 d. This indicates a slow and unselective degradation of **10** in solution.

Figure S17: $^{13}\text{C}\{^1\text{H}\}$ NMR spectra of the reaction of **10** with ^{13}CO in CD_2Cl_2 before the addition of ^{13}CO , after 30 min, 24h, 7d, 9 d and 10 d.”

“

Figure S29: ¹H (top) and ¹³C{¹H} (bottom) NMR spectra of **12** in CD₂Cl₂ (*: solvent).”

Minor

2. line 106 – add a range of C-Bi-C and Bi-Bi-C bond angles for Ph₄Bi₂ and Mes₃Bi for comparison to those in compounds 5-7.

REPLY: Ranges of bond angles for Ph₄Bi₂ and Mes₃Bi have been added. The revised version of the relevant section in the manuscript reads as follows:

“The bismuth atoms do not show any directional bonding interactions towards the [K(crypt)]⁺ cations based on distance criteria. They are found in a bent coordination geometry (C–Bi–C, 92.36(18)–97.88(16)°), with Bi–C bond lengths of 2.259(5) to 2.312(5) Å. These values are close to those reported for the corresponding dibismuthanes (e.g. Ph₄Bi₂: Bi–C, 2.26(2)/2.28(2) Å; C–Bi–C/Bi,

90.9(5)–98.3°⁴⁹ and triaryl bismuthanes (e.g. Me_3Bi : Bi–C, 2.31(1)–2.32(1) Å; C–Bi–C, 94.7(4)–107.6(4)°).⁵⁰ The close similarities in Bi–C bond lengths between these compounds speak against significant Bi–C multiple bond character, hinting at a considerable charge localization at the bismuth atoms of the bismuthides.”

3. lines 123-127 – Comment on why the carbonyl ¹³C signals show significant low field shift verses those of lighter group 15 acyl compounds.

REPLY: It has to be kept in mind that the substitution patterns at the pnictogen atoms in the bismuth compounds and the literature-known N and P compounds differ. Nevertheless, we agree that these considerable differences in ¹³C NMR spectroscopic chemical shifts should be discussed. In view of the ineffective orbital overlap between the n(Bi) and the $\pi^*(\text{C}=\text{O})$ orbitals, a higher chemical shift for the resonance of the carbonyl group in the ¹³C NMR spectrum of the bismuth species can be expected. The relevant section of the manuscript has been rephrased as follows:

“These signals show a considerable low-field shift when compared to those of lighter group 15 acyl compounds, e.g. $\text{Me}_2\text{NC}(\text{O})\text{Ad}$ (177.0 ppm)⁵² or $\text{Ter}i\text{PrPC}(\text{O})\text{Ad}$ (226.4 ppm, Ter = 2,6-bis-(2,4,6-trimethylphenyl)phenyl),⁵³ which is ascribed to inefficient $n(\text{Bi})\rightarrow\pi^*(\text{CO})$ interactions due to poor orbital overlap in **9** and **10**, as compared to the lighter pnictogen species.”

4. lines 133-135 – Comment on why the C=O stretching frequencies of 9 and 10 are higher than those of acyl phosphane and arsane compounds.

REPLY: We thank reviewer 1 for pointing out this omission. In these compounds, donation of electron density from a lone pair at the pnictogen atom to the antibonding $\pi^*(\text{C}=\text{O})$ orbital will weaken the C=O bond, thereby decrease the C=O stretching frequency. This $n(\text{Pn})\rightarrow\pi^*(\text{C}=\text{O})$ interaction becomes less efficient as the principal quantum number of the pnictogen atom Pn increases (due to less efficient orbital overlap). We agree that it will be relevant to large sections of the readership to briefly point out this interpretation. We have thus revised the relevant section of the manuscript, following the reviewer’s suggestion:

“In IR spectra of compounds **9** and **10**, the C=O stretching bands were found at wave numbers of 1691 and 1690 cm^{-1} , which are slightly larger than those reported for the acyl phosphane $i\text{PrTerPC}(\text{O})\text{Ad}$ (1653 cm^{-1})⁵³ and the acyl arsane $\text{TMS}_2\text{AsC}(\text{O})t\text{Bu}$ (1667 cm^{-1}).⁵⁴ Again, this is in agreement with inefficient $n(\text{Bi})\rightarrow\pi^*(\text{C}=\text{O})$ interactions compared to the analogous interactions in the P and As species.”

Reviewer #2 (Remarks to the Author):

The work by Lichtenberg's group describes the reversible carbon monoxide extrusion/insertion at a Bi(III) center in bismuthanes. This study represents a valuable addition to the growing field of CO activation by main group elements, which can mimic the behavior of transition metals. While CO activation by main group species has been demonstrated in low-valent compounds, multiple-bonded species, geometrically constrained systems, and frustrated Lewis pairs, a detailed understanding of the elementary steps and model systems that exhibit reversible CO transformations remains fundamentally important for advancing selective and competitive CO-based reactivity.

In this context, reversible CO insertion/extrusion has only been reported in a limited number of cases involving boron and aluminum, with one photochemical example for germanium. The present discovery of reversible CO extrusion/insertion at a Bi center in bismuth acyl compounds significantly expands the scope of this chemistry. The novelty and conceptual advance presented in this work are suitable for publication in Nature Communications.

The experimental work is thorough, with careful characterization of the synthesized compounds by NMR, IR, UV-Vis, HRMS, single-crystal XRD, and elemental analysis. The electronic properties of key compounds are also well investigated by DFT calculations. The central conclusion—reversible CO extrusion/insertion—is strongly supported by experiments, including those using ¹³C-labeled CO. While the light-induced CO extrusion is clearly demonstrated, there remains some ambiguity regarding the role of light in the insertion process.

REPLY: We thank reviewer 2 for the evaluation of our manuscript and appreciation of our work. All his/her comments have been addressed in detail as follows:

Major Comments

*--Since the reversible CO insertion/extrusion is only observed for **9** featuring a cyclic bismepine moiety, while **10** bearing more general aryl groups does not exhibit such reactivity, the depiction in the right panel of Figure 1c should be revised accordingly to accurately reflect this limitation.*

REPLY: We are grateful for the careful observation and agree with reviewer 2 on this point. In view of the new results added in the course of the revision (reversible (albeit less selective) CO extrusion/insertion not only in **9/11**, but also in **10** and its newly added counterpart **12**), we feel like the right panel of Figure 1c could remain as shown in the original version of the manuscript, if the reviewer and the editor do not oppose.

*--The reaction of **11** with CO under ambient light at room temperature is extremely slow, yielding only ~5% of acyl bismuthane **9** after 28 days. Does this suggest that the formation of **9** is thermodynamically unfavorable or that the kinetic barrier is prohibitively high? A discussion in the main text would be appropriate. Additionally, did the authors attempt this reaction under elevated temperatures or with LED light irradiation to accelerate the conversion? Is the CO insertion thermally promoted?*

REPLY: We thank reviewer 2 for these important comments. The previous wording of the relevant section in the supplementary information might have been misleading. After 4 d approximately 6% of acyl bismuthane **9** are formed and the yield of **9** did not increase further over a total period of 28 d. In addition, heating samples of **11** under an atmosphere of CO did not affect the outcome of the reaction

in terms of the equilibrium ratio **9/11** (albeit tested in the small temperature window that is offered by CH₂Cl₂, the solvent that was favorable for the entire set of experiments). When light was excluded from the sample (as well as possible with long-term experiments, sample preparation, and reaction monitoring), the reaction proceeded at a significantly lower rate, suggesting an important role of light in the transformation. It has to be noted however, that the reverse reaction is also light-promoted. When discussing the role of light, it also has to be taken into account that the irradiation of **9** with an LED, leads to rapid formation of **11** along with up to 20% of side-products (as compared to a slower but more selective reaction under ambient light), creating an overall complex scenario. When it comes to segregating the influence of thermodynamic and kinetic parameters of the reaction, the formation of **11** appears to be thermodynamically favored according to the experimental findings. There is no strong preference however, so that **9** and **11** can be observed in equilibrium under given conditions. This is also supported by a calculated Gibbs energy close to zero for the reaction of **9** to give **11** and CO. The time / conversion data of the reaction suggest that the kinetic barrier is significant, but can be overcome to realize an equilibrium scenario.

The following sections have been modified to describe our findings more precisely and to incorporate the results of additional experiments that have been performed inspired by this reviewer's comments.

Main part:

"The addition of CO (1.0 bar) to a solution of **11** in CD₂Cl₂ led to the formation of minor amounts (ca. 6%) of **9** after 4 d, according to ¹H NMR spectroscopy indicating a partial reversibility and suggesting a thermodynamic equilibrium. Longer reaction times, higher temperatures (up to 40 °C), or Lewis basic solvents such as THF did not increase the yield considerably, while exclusion of light impeded the reaction."

Supplementary Information:

"In a J. Young NMR tube, **11** (13 mg, 25 μmol) was dissolved in CD₂Cl₂ (0.5 mL). The atmosphere above the solution was exchanged with CO (1.0 bar) *via* three freeze/pump/thaw cycles. The reaction was monitored via ¹H NMR spectroscopy and the sample stored at room temperature under ambient light between each measurement. Over the course of 4 d the formation of small amounts of acyl bismuthane **9** (up to approximately 6%), along with the formation of traces of other unidentified products, was observed and the amount of **9** in solution remained constant afterwards (monitored for up to 28 d). Similar results were obtained when other solvents (e.g. THF-*d*₈) were used and when the reaction was performed at elevated temperatures (up to 40 °C in CD₂Cl₂). Irradiation of a sample in CD₂Cl₂ that had reached equilibrium (i.e. 6% of compound **9**) with a blue LED (λ_{max} = 460 nm) for 1 h led to a decrease of the amount of **9** in solution (from 6% to 4%), concomitant with the formation of a small amount of dibismuthane **1** (approximately 3%) according to ¹H NMR spectroscopy. The formation of similar amounts of dibismuthane **1** as a side-product was also observed during the irradiation of a solution of **9** in CD₂Cl₂ (under an atmosphere of argon, not of CO), as described in the main part."

--Although the light-driven CO extrusion from 9 to 11 is well supported by experiments, additional control studies are necessary to confirm whether light is also required for the reverse CO insertion (from 11 to 9). This key aspect is not discussed in the main text. The authors are encouraged to include further control experiments, possibly in the Supporting Information, to strengthen their mechanistic claims.

REPLY: Performing the reaction between **11** and CO in the dark led to a significant decrease of the rate of the reaction. After prolonged reaction times, the ratio of **11/9** reached the same value as observed in reactions that were carried out under ambient light. When evaluating these experiments, it had to be kept in mind that only small amounts of the CO-insertion product are formed, long reaction times are necessary and the exclusion of light in the strictest sense is not possible, because the light shield has to be removed for the insertion of the sample into the NMR spectrometer, for instance. Nevertheless, a significant effect of light on the outcome of the experiment could be noted. These insights have been added to the revised version of the Supplementary Information:

“When a sample was prepared in the same way as described above (i.e. 10 mg **11** in CD₂Cl₂, 1.0 bar CO), but then stored under exclusion of light between ¹H NMR spectroscopic measurements (as far as possible), the slow formation of minor amounts of **9** was still observed. However, the formation of **9** was significantly slower (e.g.: 1% **9** after 24 h in the absence of light and 3% **9** after 3 h in the presence of light). In the absence of light (the reaction vessel was protected by a cover, which had to be removed for the insertion into the NMR spectrometer), the spectroscopic yield of **9** reached approximately 6% after 13 d and remained constant afterwards. These findings indicate the acceleration of the reaction under photochemical conditions (in this context, it should be noted that UV radiation from ambient light may contribute to the activation of **11** that is exposed to CO).”

Minor Comments

*--Exploring potential reactivity of **9** would substantially enhance the impact of this work. For example, has the reactivity of **9** toward small molecules such as H₂ been investigated?*

REPLY: We thank the reviewer for this suggestion and have carried out experiments along these lines. However, when a solution of **9** was stored under an atmosphere of H₂, for instance, only the CO extrusion with formation of **11** was observed via ¹H NMR spectroscopy. While compound **9** does not seem to react with H₂, we certainly intend to further explore the reactivity of the compounds described in this manuscript towards small molecules and related substrates in the future.

--Page 2, line 62: Please be cautious with the phrase “the first set of isolable diaryl bismuthides.” Although prior literature may not include fully characterized examples, the generation and transformation of diaryl bismuthides as reactive intermediates, and the crystal structure of [K(crypt)][BiPh₂], have been reported (Refs 40–41). While the authors present a new synthetic route and thorough characterization, the claim of “first” is overstated.

REPLY: As reviewer 2 describes, previous work has described the *in situ* generation and transformation of diaryl bismuthides. Additionally, the crystal structure of [K(crypt)][BiPh₂] (without a yield or additional analytical data) has been reported. These works are cited in Refs. 37, 38, 40 and 41. We fully acknowledge that diaryl bismuthides have been described before (with one XRD analysis as the only means of direct characterization). We meant to use the wording “first set of isolable diaryl bismuthides” (with the restrictions “set” and “isolable”) to communicate this situation. In order to clarify our focus of having isolated and characterized a number of diaryl bismuthides, we have slightly rephrased the relevant section according to the reviewer’s comment. In the revised version of the

manuscript, we now use the phrase “the first set of isolable and thoroughly characterized diaryl bismuthides”.

--Page 2, line 63: The term “unprecedented inverse-polarity approach” should be reconsidered. Inverse-polarity reactions involving diaryl bismuthides and electrophiles have already been documented (Refs 38, 40). In this work, the electrophile is an acyl chloride. Thus, the use of “unprecedented” appears exaggerated.

REPLY: We avoid the word unprecedented in the revised version of the manuscript, restricting the phrase to “inverse-polarity approach”.

--Page 3, line 78: The term “elusive” to describe bismuthides is misleading and should be revised.

REPLY: A combined answer is given in our reply to the reviewer’s comment on “page 3, line 75” (see below).

--Page 3, line 79: The statement “But the rational synthesis, isolation, and characterization of these compounds have not been reported to date” should be modified. Previous work by Suzuki and others (Ref 40) should not be overlooked.

REPLY: A combined answer is given in our reply to the reviewer’s comment on “page 3, line 75” (see below).

--Page 3, line 75: The phrase “without stabilizing electronic effects” is problematic. Any scaffold or substituent inherently exerts some electronic influence. Please revise this statement for accuracy.

REPLY: The term elusive was meant to be connected to diaryl bismuthides. Furthermore, we meant to communicate that diaryl bismuthides have not been isolated and properly characterized to date and we are convinced that the term elusive is suitable for compounds which could so far only be generated in situ and for which one single-crystal analysis is the only available analytical data so far. We understand that in the original phrasing, this statement might have been connected to the entire class of bismuthides, which was not intended. In addition, we fully agree that a number of simple bismuthides (including a small number of diaryl species) has previously been generated in situ and we clearly want to acknowledge these contributions. Nevertheless, it is important to point out that these synthetic approaches did not allow for the isolation of such compounds.

We fully agree that any scaffold exerts some electronic influence, but this is not necessarily a stabilizing influence. In the cases we cite, the aromatic nature of the bismuthides or the α -silyl effect have been exploited to stabilize the bismuthides through delocalization of electron density. This is a valuable strategy, but restricts the accessible compounds. We meant to communicate that the compounds we report do not rely on the application of this strategy. Thus, we have rephrased the relevant section as follows in order to meet the criticism raised by reviewer 2 and to avoid any confusion:

“The *in-situ* generation of simple bismuthides $[\text{BiR}_2]^-$ has been suggested based on trapping experiments (R = aryl, alkyl).^{30–36} These parent compounds do not benefit from stabilizing electronic effects (such as aromatization or the α -silyl effect)^{37–40} that aim at significant delocalization of electron

density. One set of single-crystal X-ray diffraction data of a diphenyl bismuthide (without additional analytical data or a yield) has serendipitously been reported,⁴¹ granting first glimpses at the yet elusive class of diaryl bismuthides. But a rational synthesis facilitating the isolation and characterization of these simple bismuthides has not been reported to date.”

--Figure 4d: Consider removing the equilibrium arrow between 9 and 9' to better illustrate the transformation pathway between 9, 11, and 9'.

REPLY: We thank reviewer 2 for this suggestion. We originally used the equilibrium arrow here because the CO exchange could in principle also proceed without the formation of **11**. So it was meant to be a cautious representation covering mechanistic possibilities that cannot be strictly ruled out. With our new data on compounds **10** and **12**, we like to keep the scheme in its original form, because the CO exchange in these species is quicker than CO elimination. Of course, CO exchange could still involve **12** in a rapid equilibrium reaction, but we think it is better at this stage not to strictly rule out a scenario in which **9** and **9'** or **10** and **10'** can interconvert without the formation of **11** or **12** (e.g. via a transition state with bridging CO ligands, just to suggest one among many possibilities).

--Could kinetic data—such as apparent rate constants, activation energy, and activation parameters (apparent rate constants, activation energy, enthalpy and entropy of activation)—be extracted from the ¹³CO labeling experiments involving 9? Such information would provide a deeper understanding of the reaction mechanism.

REPLY: We fully agree that kinetic data could provide a deeper understanding of the reaction. Such data could in theory nicely be extracted from the ¹³CO labeling experiments. In practice, however, we refrain from doing after a careful analysis, because several reactions take place in parallel in our specific case. Firstly, CO exchange between **9** and **9'** takes place and is the subject of investigation. In addition, however, **9** also undergoes CO extrusion to give **11** and **11** reacts with CO to give **9** and with ¹³CO to give **9'**. Furthermore, minor amounts of side products form in the reactions generating an additional source of uncertainty for the determination of kinetic data. Overall, considering the number of involved reactions and associated uncertainties, kinetic data from this experiment would be connected to large errors. Therefore, we refrain from reporting rate constants and activation parameters in this scenario.

--In the Supporting Information, the mention of compound 3 seems out of place and possibly erroneous:

--Table S5: Data obtained from ¹H and ¹³C{¹H} NMR spectra of reaction of 3 with ¹³CO in CD₂Cl₂.

--When the same reaction was performed and the sample stored under exclusion of light between measurements (as far as possible), no signs for CO extrusion or the incorporation of ¹³CO into 3 were observed, giving further evidence of the light-driven nature of the reaction.

--Please clarify whether this section refers to 3 correctly or if it should refer to 9.

REPLY: We thank the reviewer for the careful evaluation of our work and for spotting the erroneous mentioning of compound **3**. These sections were meant to refer to compound **9** and have been changed accordingly.

--Figure S17: The peak at 146.1 ppm is not visible. Please zoom in on this region of the ^{13}C NMR spectrum for clarity.

REPLY: We have added an enlargement of this region of the spectrum and the corresponding region of the $^{13}\text{C}/^1\text{H}$ HMBC spectrum to the relevant figure (this was Figure S17 in the original SI and is now Figure S21 in the revised version of the SI). To further clarify that the resonance was also detected in HMBC NMR experiments we have added the following sentence to the figure caption: *“The presence of the weak resonance signal at 146.15 ppm in the $^{13}\text{C}\{^1\text{H}\}$ NMR spectrum was additionally confirmed via $^{13}\text{C}/^1\text{H}$ HMBC experiments (upper black box, correlation peaks involving C2/13 are highlighted in light blue).”*

Reviewer #3 (Remarks to the Author):

REPLY: We thank reviewer 3 for the evaluation of our work and the time he or she invested to co-review the manuscript.